# Wilhelm von Waldeyer: Important Steps in Neural Theory, Anatomy and Citology

**DOI:** 10.3390/brainsci12020224

**Published:** 2022-02-06

**Authors:** Vicentiu Mircea Saceleanu, Aurel George Mohan, Razvan Adrian Covache-Busuioc, Horia Petre Costin, Alexandru Vlad Ciurea

**Affiliations:** 1Neurosurgery Department, Sibiu County Emergency Hospital, 550245 Sibiu, Romania; vicentiu.saceleanu@gmail.com; 2Neurosurgery at “Lucian Blaga” University of Medicine, 550024 Sibiu, Romania; 3Department of Neurosurgery, Bihor County Emergency Clinical Hospital, 410167 Oradea, Romania; 4Neurosurgery, Faculty of Medicine, Oradea University, 410610 Oradea, Romania; 5General Medicine, “Carol Davila” University of Medicine and Pharmacy, 020021 Bucharest, Romania; razvan_adrian69@yahoo.ro (R.A.C.-B.); horiacostin2001@yahoo.com (H.P.C.); 6Neurosurgery, “Carol Davila” University of Medicine and Pharmacy, 020021 Bucharest, Romania; prof.avciurea@gmail.com; 7Neurosurgery Department and Scientific Director at the Sanador Clinical Hospital, 70000 Bucharest, Romania

**Keywords:** neuron theory, Waldeyer history, Waldeyer citology, Waldeyer anatomy

## Abstract

Heinrich Wilhelm Gottfried von Waldeyer-Harz is regarded as a significant anatomist who helped the entire medical world to discover and develop new techniques in order to improve patient treatment as well as decrease death rates. He discovered fascia propria recti in 1899, which is important in total mesorectal excision which improves cancer treatment as well as outcomes. He played an important role in developing the neuron theory which states that the nervous system consists of multiple individual cells, called neurons, which currently stands as the basis of the impulse transmission of neurons. Waldeyer was also interested in cytology, where he made a substantial contribution, being the first who adopted the name “Chromosome”. Therefore, he accelerated the progress of what it is now known as Genetics. In conclusion, starting from the Fascia propria recti and continuing with great discoveries in cytology and neuron theory, Wilhelm von Waldeyer represents a key person in what we today call medicine.

## 1. Introduction

Heinrich Wilhelm Gottfried von Waldeyer-Hartz (Figure 1) was a well-known German anatomist who is known for his efforts in giving both the medical world and humanity an acquaintance in multiple fields of study such as anatomy, embryology and pathology, which today plays a vital role in treating genetic diseases and cancer. 

He was born on 6 October 1836 in Hehlen, a small village near Brunswick. He completed his studies at the Gymnasium Theodorianum in Paderborn, were he obtained a graduation diploma in 1856, which attested his eligibility of attending university courses. Further on, he attended the Universität Göttingen were he focused his studies on mathematics and natural sciences [1]. 

This is a place that played an important role in the future of Waldeyer, because he met the recognised anatomist Jakob Henle (1809–1885), who discovered the loop of Henle which has great importance in the kidney’s physiology. Waldeyer was so impressed with Henle’s work that he entered medical school in 1857. In 1861, he acquired his doctorate diploma based on his thesis entitled “De claviculae articulis e functione” [2]. 

Later on, in 1867, he became a professor of pathological anatomy in Breslau, then Waldeyer became a full professor in Strasbourg in 1872, and in 1883, he moved to Berlin where he lived for more than 33 years working at the Institute of Anatomy [3].

Before his death on 23 January 1921, his desire was for his hand, skull and brain to be preserved at the Institute of Anatomy in Berlin in order to be studied and examined. Hans Virchow was the one who dissected the hands and published an entire detailed description of the anatomy of this body part of Waldeyer (Figure 2) [4]. However, the studies based on his brain and skull were not assigned to Virchow and were not found. The entire idea behind his desire to donate his body parts to the Institute was a popular decision among well-known people from the medical world of that era due to a belief that distinguished signs could be seen in tremendously intelligent people’s brains [2].

## 2. Waldeyer’s Medical Contributions

One of Waldeyer’s biggest anatomical contribution was represented by the “Fascia propria recti”, which at the time of discovery in 1899, did not present such a great medical interest, but in the last century, its importance in surgical practice has grown higher and higher due to its importance in total mesorectal excision, a surgical procedure involved in rectal cancer treatment. It was described for the first time by Professor Bill Heald in 1982, at the Basingstoke District Hospital in the United Kingdom [5].

Fascia propria recti is regarded as a thin layer of connective tissue which lies between the presacral fascia and the rectal proper fascia. It is also known as the rectosacral fascia, according to its position, defining the retrorectal space in two compartments: a superior and an inferior one [6].

The great debate that appeared around the fascia propria recti is whether Waldeyer was the first one who discovered it or not. In the first edition of the anatomical book “Traité d’Anatomie Humaine”, revised by P. Poirier, which was probably published in 1894, but definitely between 1892 and 1896 in Paris, Toma Ionescu described this fascia for the first time, under the name of “rectal sheath”, about 5 years before the name of “fascia propria recti” was spread around the entire medical world [7].

It is not clear why Toma Ionescu was not perceived as the first anatomist who discovered it, but some probable theories suggest that it was mostly because of the big difference between Toma Ionescu and Waldeyer’s age. Waldeyer was at that time with 25 years older than Toma Ionescu and was already one of the most well-known anatomists across the world, having a wider influence and a recognised reputation [7].

Nevertheless, French authors carry the entire merit of giving Toma Ionescu the credits for his discovery, considering him as the first one who claimed the name for the rectosacral fascia.

## 3. The Neuron Theory

The neuron theory, which is also called the neuron doctrine, represents an idea of Santiago Ramón y Cajal, which states that the nervous system consists of multiple individual cells called “neurons”, which have an individual structure and function, working together in order to create a singular and refined machinery that controls the entire human body [8].

However, the path to achieve this concept was not an easy one and Wilhelm von Waldeyer played an important role in expressing the neuron theory.

The history of the neuron theory starts back in 1873, when Camillo Golgi invented a new staining method, known as “la reazione nera” (“black reaction”), later called Golgi staining technique in his honour [9].

This method was used for microscopic research, which at that time was difficult due to the lack of staining techniques. Therefore, the new method discovered by Golgi played a vital role in the discovery of the nervous system, because he could differentiate the dendrites from the axon of the neuron.

Thus, he observed an entire network of neurons in the grey matter and proposed what was called “The reticular theory”. This concept proposed that the entire cerebro-spinal axis was one continuous neural network that acted as a single organ. This theory represented the main idea of how the entire nervous system works, but in reality, the truth was totally different from what Camillo Golgi proposed [10].

In 1887, Santiago Ramón y Cajal used the Golgi staining technique to study the neural network, making a discovery that would change the entire approach of the nervous system (Figure 3). He discovered that between the neurons, there is not a continuous link, but instead there is a space between them, which is now known as the synaptic cleft. This was the moment “The neuron theory” was born [11].

Golgi’s concept of a continuous nervous system was therefore obsolete, and even though he never agreed with Cajal’s theory, Waldeyer was a firm supporter of it. Moreover, the impact of Waldeyer’s contributions is mostly represented by naming the nervous cells “neurons”, which comes from the Greek word “sinew” [12].

Therefore, the neuron theory constituted a solid base for the following discoveries in terms of impulse transmission, as well as structural and functional particularities of the neurons, which could be later described using electronic microscopy in order to make a clear statement about the way the entire nervous system works [13].

However, we do not have to assume that Golgi’s reticular theory was entirely wrong. Nowadays, studies have determined that there is an intense interconnectedness between neurons and astrocytes, and even if we could describe the nervous system as a network composed of many independent cells, it is much more important to assume that it works as a unitary and perfectly coordinated system.

## 4. Waldeyer’s Contributions to Cytology

Cytology in the 19th century represented a controversial study subject due to the lack of information, as well as the absence of the lab techniques needed in order to analyse the structure of the cell and its mechanisms. However, different studies were conducted and step by step, the researchers of that time made rapid progress. Waldeyer, played an important role in refining the cytology.

In 1888, Waldeyer published an article intitled “Über Karyokinese und ihre Beziehungen zu den Befruchtungsvorgängen” (“About karyokinesis and its relationships with the fertilization processes”) [14], which signifies what is now called a review article, since it has 210 references, used to objectify a vast amount of information in just one paper.

Among the scientists of that century who were citated in this extended review, names including Rudolf Virchow, Theodor Boveri, Oskar Hertwig, Edouard-Gerard Balbiani, Walther Flemming and many others provided both theoretical and experimental information which was used to enhance the explanation of the entire fertilization and karyokinesis process.

One of the most relevant information that can be extracted from this article, is the word “Chromosomen” (in German) [14], which was translated into English under the form we use today of “Chromosome”. Before this name was introduced by Wilhelm von Waldeyer, the name “Chromatinelemente” (Chromatic elements) was proposed by Theodor Boveri and used by the entire scientific community [2].

However, there was a long way that had to be followed in order to reach the chromosome discovery. First of all, near the middle of the 19th century, Theodor Schwann (1810–1882) and Matthias Schleiden (1804–1881) were regarded as the discoverers of cell theory, suggested in 1838–1839. They strongly believed that the cells were produced de novo from a substance called “cytoblastem”, which did not have a specific structure [15].

Even though this theory sounds aberrating these days, the scientific community did not have any strong experimental or theoretical information to correlate, and this lack of data led to a wrong perception of the cytokinesis.

The theory was not rejected until 1855, when Rudolf Virchow (1821–1902) stated “omnis cellula e cellula” [15], which means that every new cell is created from a pre-existing one through division (Figure 4).

Nevertheless, after Virchow’s statement became clear for the entire scientific community, the new debate about nucleus division became more and more thought-provoking for all researchers. It could be seen at that time that during cell division, the nucleus disappears and appears immediately along with the birth of new daughter cells.

Some of the scientists of that time considered that karyokinesis was actually a “generatio spontanea” inside the cells, while others, such as Walther Flemming, were supporters of the indirect division (amitosis) of the nucleus. Later on in 1917, Oscar Hertwig made a discovery that the role of the chromosomes is represented by their hereditary information, helping the cells to become specialised according to the indications made by the chromosomes [16].

Therefore, the 19th century represented a century of discovery that laid the foundation for future research. Starting from cell theory and going up to the main role of chromosomes in deciding the fate of the cell, many experimental advancements were made and opened a new path in understanding the basic functions of the cell.

Moreover, Waldeyer represented a key person in the development of the cytology, and in addition to familiarising the name “chromosomes” throughout the world, his exceptional microscopy methods led to important observations in fertilization and karyokinesis, being able to describe even the way polar bodies are formed during oogenesis.

## Figures and Tables

**Figure 1 brainsci-12-00224-f001:**
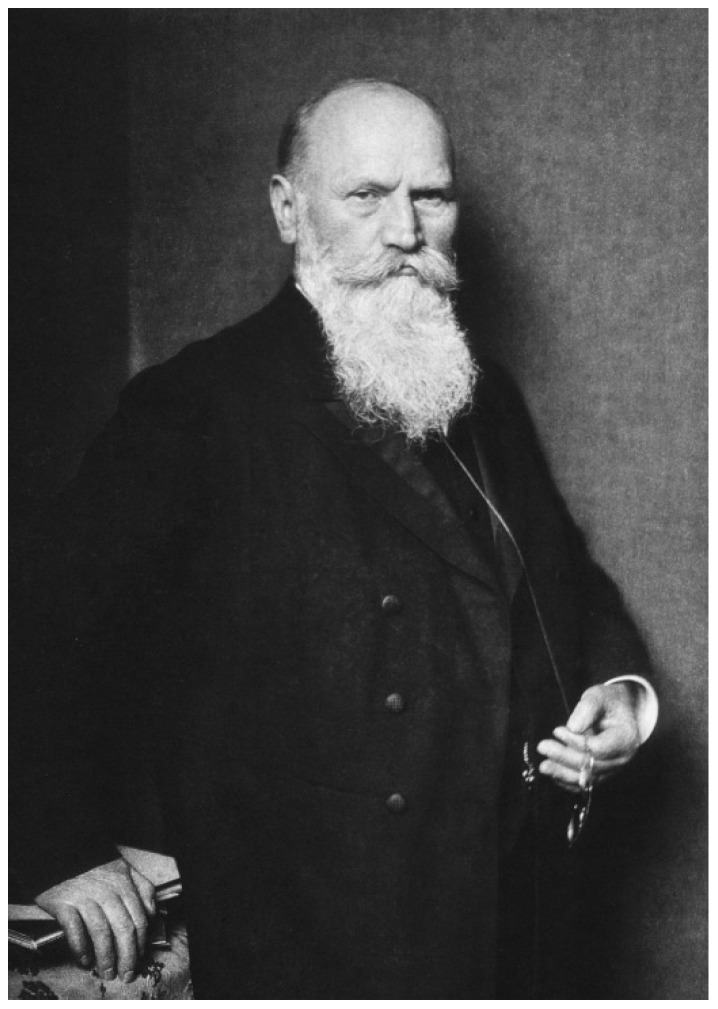
Heinrich Wilhelm Gottfried von Waldeyer-Hartz (1836–1921).

**Figure 2 brainsci-12-00224-f002:**
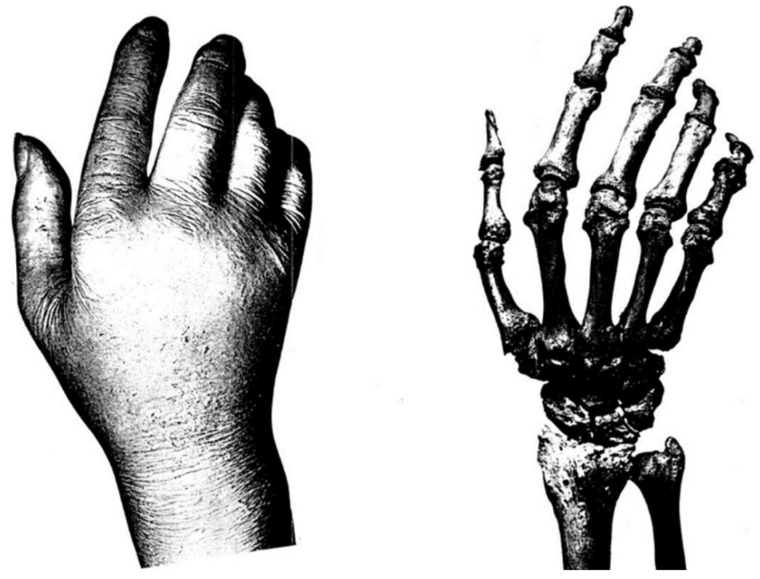
Waldeyer’s right hand (**left**) and the skeleton of the right hand (**right**) according to Hans Virchow’s description (1923) [4].

**Figure 3 brainsci-12-00224-f003:**
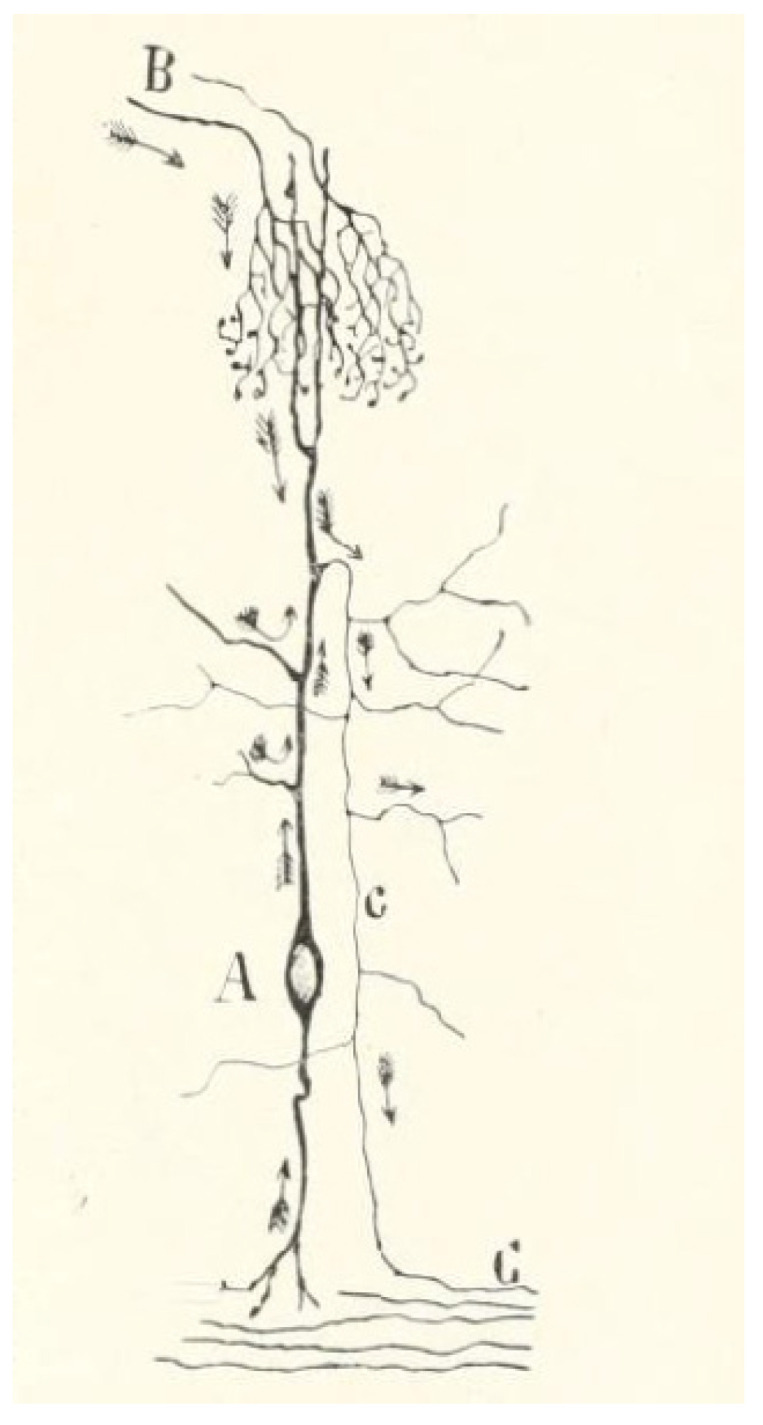
A graphic description of a synapse involved in vision (optic lobe), according to Cajal (1909), using the Golgi staining method. The arrows indicate the direction of the nervous impulse. (**A**) neural body, (**B**) afferent fibres. (**C**) axonal fibre [11].

**Figure 4 brainsci-12-00224-f004:**
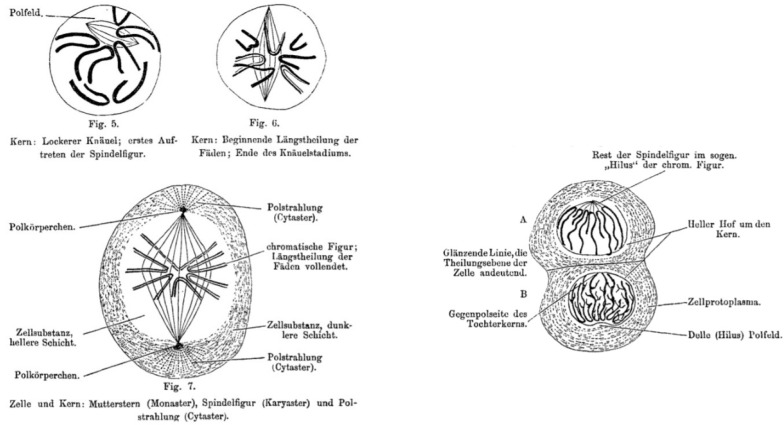
Wilhelm von Waldeyer’s representation of cytokinesis and karyokinesis, according to the extensive review entitled “Über Karyokinese und ihre Beziehungen zu den Befruchtungsvorgängen” [14]. Figures 5–7 in ref. [14].

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
