# Peer review of "Wilhelm von Waldeyer: Important Steps in Neural Theory, Anatomy and Citology"

_brainsci, 2022, doi:10.3390/brainsci12020224_

Round 1

Reviewer 1 Report

Comments and Suggestions for Authors

The review is interestingly written and it is very important that in the modern transient circumstances of the development of science, it reminds of its roots. Unfortunately, there seems to be an important flaw in the text.

Major:

  1. Judging by the name, it is dedicated not only to the memory of an outstanding scientist, but also to the 130th anniversary of the life of the term "neuron" in brain science. That is, it projects ideas and discoveries into our time. However, the content of the meaning of the "neural theory" after 130 years is revealed very superficially. In recent decades, the main content of neuronal theory has shifted to the field of heterochemistry of nerve cells, the fundamental importance of the chemical diversity of neurons, and the participation of the genome in the functions of nerve cells.

I believe that the reason is that the authors are more professionals in the field of medicine than neurobiology.

It is recommended either to adequately fill this gap, or to clarify the name.

Minor:

About the story, you can add:

  1. As for biography of Heinrich Wilhelm Gottfried Waldeyer. He was Foreign Corresponding Member of the St. Petersburg Academy of Sciences (1894).

[Profile of Waldeyer Heinrich Wilhelm Gottfried on the official website of the RAS:

http://www.ras.ru/win/db/show_per.asp?P=.id-49789.ln-ru   ]

As for history of term “neuron”:

  1. On September 19, Jan Evangelista Purkyně at the Congress of German Naturalists and Physicians in the Karolinum in Prague 1837 presented his discovery of microscopic objects in cerebellum, which later makes him eponym - Purkinje cells. The most famous images of Pc are on pictures by Santiago Ramón y Cajal.

Minor comments are advisory in nature, at the discretion of the authors.

Author Response

Thank you for providing your expertise regarding the review of our manuscript!

We took into consideration your suggestion for a title change and came up with "Wilhelm von Waldeyer: Important steps in neural theory, anatomy and citology" - we think this title suits the content of our manuscript better and it will clarify the name.

Best regards and our kindest wishes!

Reviewer 2 Report

The majority of comments are minor. Nice manuscipt on someone that greatly impacted science.

Line 3 & 4-check author instructions as whether assistant, associate, and professor should be included in co-author list, I don’t think so.

Line 19 “which currently states” is awkward, please rephase

Line 20 suggest “where he made a substantial contribution”

Line 33 the legend has the incorrect dates of life should be 1836-1921

Line 41 “is he assigned the medical school” is awkward rephase perhaps entered medical school

Line47 & 52 replace “will” with “desire” (will has multiple meanings) desire conveys the thought better

Line 70 replace “is” with “was”

Line 130 rapid “progress” not progresses

Line 176 replace leaded with led

For Figures, the sources are referenced which may be all that is required, but check with instructions to authors as to whether permissions to use the figures from published work (just verify it is not required)

Author Response

Thank you for providing your expertise regarding the review of our manuscript!

We took into consideration all of your comments and suggestions and made all the modifications accordingly.

Best regards and our kindest wishes!